# Birds of a Feather Trust Together: Knowing When to Trust a Classifier via Adaptive Neighborhood Aggregation

**Miao Xiong[1], Shen Li[1], Wenjie Feng[1], Ailin Deng[2], Jihai Zhang[2], Bryan Hooi[1,2]**
*{miao.xiong, shen.li}@u.nus.edu, wenjie.feng@nus.edu.sg, ailin@u.nus.edu, {jihai, dcsbhk}@comp.nus.edu.sg*
[1] *Institute of Data Science, National University of Singapore*
[2] *Department of Computer Science, National University of Singapore*

**Reviewed on OpenReview:** *https://openreview.net/forum?id=p5V8P2J61u*

## Abstract

How do we know when the predictions made by a classifier can be trusted? This is a fundamental problem that also has immense practical applicability, especially in safety-critical areas such as medicine and autonomous driving. The de facto approach of using the classifier's softmax outputs as a proxy for trustworthiness suffers from the over-confidence issue; while the most recent works incur problems such as additional retraining cost and accuracy versus trustworthiness trade-off. In this work, we argue that the trustworthiness of a classifier's prediction for a sample is highly associated with two factors: the sample's neighborhood information and the classifier's output. To combine the best of both worlds, we design a model-agnostic post-hoc approach NEIGHBORAGG to leverage the two essential information via an adaptive neighborhood aggregation. Theoretically, we show that NEIGHBORAGG is a generalized version of a one-hop graph convolutional network, inheriting the powerful modeling ability to capture the varying similarity between samples within each class. We also extend our approach to the closely related task of mislabel detection and provide a theoretical coverage guarantee to bound the false negative. Empirically, extensive experiments on image and tabular benchmarks verify our theory and suggest that NEIGHBORAGG outperforms other methods, achieving state-of-the-art trustworthiness performance. [1].

## 1 Introduction

In recent years, interactions with AI systems have become increasingly pervasive in all walks of our daily lives. As machine learning models become more widely involved in our decision-making processes, the robustness and trustworthiness of their decisions need to be carefully scrutinized (Varshney & Alemzadeh, 2017). This is of vital importance in many scenarios, especially in safety-critical areas, such as medical applications, where successful deployment is highly dependent on a model's ability to detect an incorrect prediction, so that humans can intervene when necessary (Shi & Jain, 2019; Chang et al., 2020; Li et al., 2021). This leads to our central question: *how can we know when the predictions made by a classifier can be trusted?*

In this paper, we investigate the trustworthiness of the prediction given by a classifier, which serves as a measure for the classifier's quality rather than the data. Concretely, given some i.i.d data and an pretrained classifier (referred to as *'base classifier'* hereinafter), the goal is to devise a discriminative and accurate *trustworthiness score*, such that **higher scores indicate a belief that the classifier's predicted class is more likely to be correct**. In this way, the users can easily determine whether they should trust the prediction output by machine learning models, or they should resort to domain experts for manual predictions. In the literature, this task is also referred to as "trustworthiness prediction", "failure prediction", or "misclassification detection" (Jiang et al., 2018; Corbière et al., 2019).

---

[1]Our code is publicly available at https://github.com/MiaoXiong2320/NeighborAgg.git.

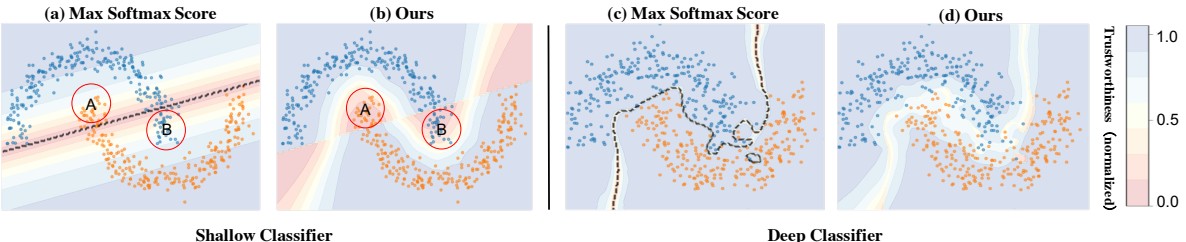

Figure 1: Comparison between max softmax scores and NEIGHBORAGG scores (ours), based on a shallow base classifier (left) and a deep base classifier (right), respectively. The color of the points indicates its ground truth label while the color of the background shows the corresponding trustworthiness score. The dotted black line demonstrates the decision boundary of the base classifier. (**a**): The points marked by circles 'A' and 'B' overstep the decision boundary, being misclassified while some of them still get high max softmax scores. On the contrary, (**b**): our algorithm can correctly assign these points the lowest trustworthiness scores. (**c**): Max softmax scores from base classifiers are potentially over-confident near the decision boundary whereas (**d**): our proposed score resolves this issue in a model-agnostic manner by inspecting their neighbors.

The most common approach is to employ a classifier's softmax output (i.e. the maximal value of a softmax vector, referred to as *confidence score* hereinafter) as the proxy for trustworthiness (Hendrycks & Gimpel, 2017). However, this approach has been found to be over-confident (Guo et al., 2017). Figure 1 illustrates this issue over 2D toy datasets: the points marked by circles 'A' and 'B' in Figure 1a are misclassified with high confidence scores. In Figure 1c, while all samples have been correctly classified, the base classifier assigns excessively high confidence scores on almost all the data points, even those near the classification boundary, making the decision boundary (full of bends and curves) prone to noise. On the contrary, our method addresses these issues by utilizing information from the *neighbors* of each point rather than just the point itself, thereby giving much lower trustworthiness scores to those misclassified points (Figure 1b) and better reflecting the uncertainty of points near the decision boundary (Figure 1d).

Other related works include uncertainty-aware methods and post-hoc methods. The uncertainty-aware methods such as MC-dropout (Gal & Ghahramani, 2016), Deep Ensemble (Lakshminarayanan et al., 2017) and Dirichlet-based approaches (Charpentier et al., 2020) typically involve retraining the classifier due to the modification of network architecture, and can incur trade-offs between classifier accuracy and the performance of trustworthiness prediction. In contrast, the *post-hoc* setting avoids such extra-cost by focusing on a pretrained classifier. Among these algorithms, Corbière et al. (2019) builds on a strong assumption that the base classifier is always over-confident, which fails in many cases (Wang et al., 2021). Trust Score (Jiang et al., 2018) leverages the neighborhood information by a hand-designed and non-trainable function, suffering from limited functional space and modeling capacity.

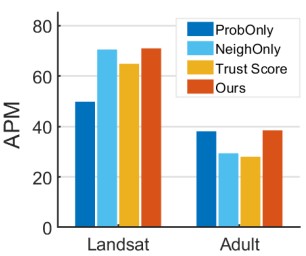

Figure 2: Sources of information for trustworthiness prediction. Our adaptive approach outperforms other methods that only use the classifier's prediction (`ProbOnly`) or neighborhood information (`NeighOnly`), and `Trust Score`.

Inspired by the commonly-held *neighborhood-homophily* assumption (Fix & Hodges, 1989), we argue that *the trustworthiness of a classifier's prediction for a given sample is highly associated with the sample's neighborhood information*, such as their labels and distances to the point itself. That is, if a sample's predicted label is consistent with the majority of its neighbors' labels, this prediction is more likely to be reliable; otherwise, we tend to assign it a lower trustworthiness score. To capture the various correlation between the sample and its neighborhood in a more flexible manner, we devise an adaptive approach to learn the scoring function, thereby ensuring superior capacity than Trust Score (Jiang et al., 2018). Figure 2 verifies the advantage by showing that the adaptive function (`NeighOnly` and `Ours`) outperforms `Trust Score`.

Furthermore, we believe that the classifier's predictive output is also an indispensable source of information for the trustworthiness prediction, if not more so than the sample's neighborhood information, particularly in cases where the classifier is sufficiently reliable or the neighborhood-homophily assumption does not perfectly meet. This is further borne out by Figure 2 where using the classifier output (`ProbOnly` and `Ours`) outperforms using only neighborhood information (`NeighOnly` and `Trust Score`) for the Adult dataset.

In this paper, we propose a model-agnostic algorithm, termed as NEIGHBORAGG, for the trustworthiness prediction by leveraging the neighborhood information and the classifier output via an adaptive scoring function that combines the best of both worlds. Theoretically, we demonstrate that our method is essentially a generalized one-hop graph convolutional network, and hence inherits the powerful modeling capacity to capture the varying similarity within each class, making it insensitive to hyperparameters for neighbor selection. Owing to the adaptive design, these two factors are able to act in a complementary manner when determining the trustworthiness score. Our method is also effective by achieving 7.63% gain on APM and 2% gain on AUC on average for the tabular dataset.

Additionally, we apply our approach to the closely related task of detecting mislabeled data samples, and propose the NEIGHBORAGG-CMD algorithm for mislabel detection. Furthermore, we obtain a theoretical coverage guarantee for this algorithm to bound the probability of false negative predictions. To the best of our knowledge, the present work is the first to adapt to real-world noisy data setting and achieves a promising result, which we believe is of independent interest.

In summary, our main contributions are as follows:

- We propose a model-agnostic post-hoc algorithm NEIGHBORAGG to measure the trustworthiness of a classifier's predictions. Moreover, by demonstrating the theoretical equivalence with a generalized graph convolutional network, we provide a better understanding into how our approach works.
- We propose NEIGHBORAGG-CMD, which adapts our method to mislabel detection and provide a noise-robust coverage guarantee to bound the false negative probability.
- Experiments on multiple tabular and image datasets showcase that the proposed NEIGHBORAGG consistently outperforms other state-of-the-art methods by clear margins. Additionally, we show that NEIGHBORAGG-CMD is able to identify mislabelled samples with promising results.

## 2 Preliminaries and Notations

We aim to measure a classifier's trustworthiness in the context of multi-class classification with $C \geq 2$ categories. Given a set of $N$ data points $\mathcal{X} = \left\{ \mathbf{x}^{(1)}, \ldots, \mathbf{x}^{(N)} \right\}$, with $\mathbf{x}^{(i)} \in \mathbb{R}^D$, and their corresponding labels $\mathcal{Y} = \left\{ y^{(1)}, \ldots, y^{(N)} \right\}$, with $y^{(i)} \in \mathcal{C} = \{0, 1, \ldots, C - 1\}$, let bold $\mathbf{y}^{(i)} \in \mathbb{R}^C$ denote the one-hot encoding of $y^{(i)}$. The dataset is split into training, validation and test set, denoted by $(\mathcal{X}_{tr}, \mathcal{Y}_{tr}), (\mathcal{X}_{val}, \mathcal{Y}_{val}), (\mathcal{X}_{ts}, \mathcal{Y}_{ts})$, respectively. Formally, we define the base classifier as a mapping $\mathcal{F} : \mathcal{X} \mapsto \mathbb{R}^C$ which takes a data point $\mathbf{x}$ as input and outputs its predicted probability vector or logits $\mathbf{p} \in \mathbb{R}^C$ and predicted class $\hat{y}$. Unless otherwise stated, vectors and matrices are denoted by boldface lowercase and uppercase letters, respectively, and sets are denoted by calligraphic letters. All vectors are treated as *column* vectors throughout the paper.

**Problem 1** (Trustworthiness Prediction). *Given a base classifier $\mathcal{F} : \mathcal{X} \mapsto \mathbb{R}^C$, the trustworthiness prediction problem is to give a trustworthiness score $t(x) = \mathcal{T}(x; \mathcal{F}, \mathcal{X}_{tr}, \mathcal{Y}_{tr}) \in \mathbb{R}$ for any $x \in \mathcal{X}_{ts}$[2], where $\mathcal{T}$ is the designed function for trustworthiness prediction, with the goal that in perfect condition:*

$$t(x) = \left\{ \begin{array}{ll} 0 & y \neq \hat{y} \\ 1 & y = \hat{y}. \end{array} \right.$$

## 3 Proposed Method

In this section, we first introduce the training and inference algorithms of our proposed NEIGHBORAGG. Then, we theoretically show that NEIGHBORAGG is a generalized version of a one-hop graph convolutional

---

[2]$\{\mathcal{X}_{ts}, \mathcal{Y}_{ts}\}$ and $\{\mathcal{X}, \mathcal{Y}\}$ are i.i.d datasets.

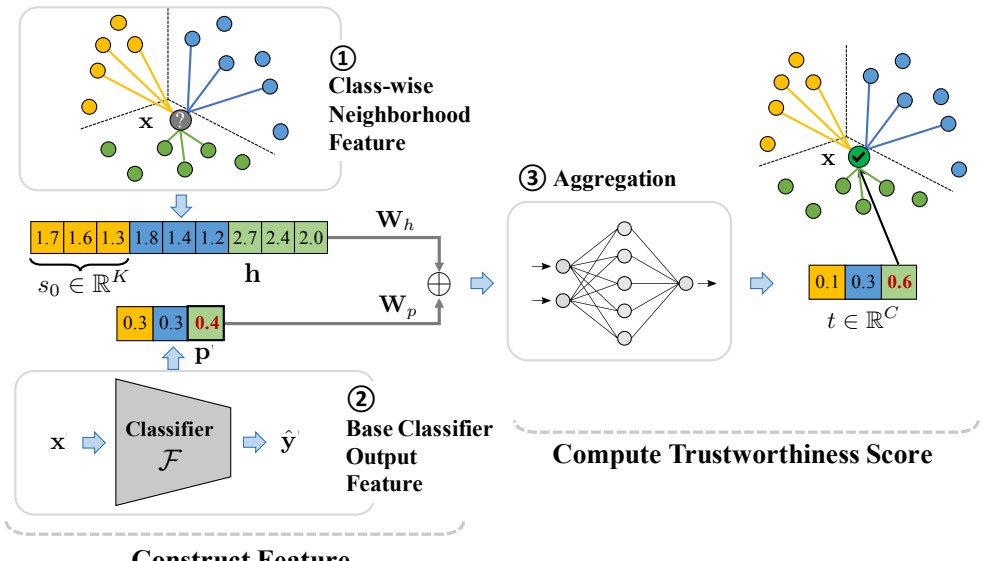

Figure 3: A conceptual illustration of the proposed NEIGHBORAGG with $K = 3$ and $C = 3$. Given a sample $\mathbf{x}$ in question, we first compute two features: (1) class-wise neighborhood feature $\mathbf{h} = [\mathbf{s}_0, \mathbf{s}_1, \mathbf{s}_2]$ that reflects the similarity to its $K$ neighbors across every class, and (2) the base classifier's probability vector $\mathbf{p}$. Then, after two linear transforms $\mathbf{W_h}, \mathbf{W_p}$ followed by concatenation, (3) NEIGHBORAGG aggregates the information and outputs the final trustworthiness score corresponding to the base classifier's predicted class. Compared to the confidence score 0.4, our approach assigns more trustworthiness to the predicted class by increasing the score to 0.6.

network. Lastly, we show a promising extension of NEIGHBORAGG applied in mislabel detection with a theoretical coverage guarantee.

## 3.1 Algorithm

As stated in the introduction, one of our key observations is that the trustworthiness of the classifier's prediction for a sample is highly associated with two information sources: *the neighborhood of the sample* and *the predictive output of the classifier*. These two components can interact in a variety of ways. How can we utilize the two information to determine a more reliable trustworthiness score?

Next, we will introduce our proposed model termed *NEIGHBORAGG* and elaborate on how these two components are constructed and efficiently aggregated by our method. The overall framework is illustrated in Figure 3.

**Feature Construction.** For a given sample $\mathbf{x} \in \mathcal{X}$, we utilize two input features: the neighborhood vector $\mathbf{h}$ and the classifier output vector $\mathbf{p}$ as shown in Figure 3.

For the classifier output feature, we use the aforementioned vector from the classifier $\mathbf{p} = \mathcal{F}(\mathbf{x})$.

The neighborhood vector consists of the similarity of a sample to its $K$ nearest neighbors across all the $C$ classes in the training dataset. Specifically, for the sample $\mathbf{x}$ and each class $c \in \mathcal{C}$, we find its $K$ nearest neighbors from class $c$ of the training dataset: $\mathcal{N}_c = \{\mathbf{n}_{c1}, \cdots, \mathbf{n}_{cK}\}$ and construct a similarity vector $\mathbf{s}_c$ as

$$\mathbf{s}_c = [s_{c1}, s_{c2}, \ldots, s_{cK}]^T, s_{ck} = \mathcal{K}_f(\mathbf{x}, \mathbf{n}_{ck}), \tag{1}$$

where $\mathcal{K}_f$ is the Laplacian kernel with a transform $f$, i.e. $\mathcal{K}_f(\mathbf{x}, \mathbf{z}) = \exp\left(-\|f(\mathbf{x}) - f(\mathbf{z})\|_2\right)$. Cosine similarity can be used as well; but we find that in practice, Euclidean distance wrapped into Laplacian kernel performs better. For tabular dataset, the transform $f$ is set to the identity mapping of the original data, which we find empirically yields sufficiently good performance. For image dataset, a more complex transform (e.g. the

---

**Algorithm 1:** Training Algorithm of NEIGHBORAGG

---

**Input:** Training set $(\mathcal{X}_{tr}, \mathcal{Y}_{tr})$; Validation set $(\mathcal{X}_{val}, \mathcal{Y}_{val})$; Base classifier $\mathcal{F}$; Kernel $\mathcal{K}_f$; Aggregator AGG;
       Training epoches $M$; Number of neighbors $K$.

**Output:** Parameters of NEIGHBORAGG: $\mathbf{W}_h$, $\mathbf{W}_p$, $\mathbf{W}$ (parameters of AGG).

**1** Initialize $\mathbf{W}_h$, $\mathbf{W}_p$, $\mathbf{W}$;

**2** **for** $c = 1$ *to* $C$ **do**

**3**     $\triangleright$ *Building class-wise KD-trees using the training set*

**4**     Split from $\mathcal{X}_{tr}$: $\mathcal{X}_c = \{\mathbf{x}|\mathbf{x} \in \mathcal{X}_{tr}, y = c\}$;

**5**     Construct a KD-tree $\text{KDT}_c$ using $\mathcal{X}_c$ based on the kernel $\mathcal{K}_f$;

**6** **end**

**7** **for** *epoch* $= 1$ *to* $M$ **do**

**8**     $\triangleright$ *Training our NEIGHBORAGG using the validation set*

**9**     **for** $\mathbf{x}$ *in* $\mathcal{X}_{val}$ **do**

**10**         **for** $c = 1$ *to* $C$ **do**

**11**             Find $K$ nearest neighbors of $\mathbf{x}$ from $\text{KDT}_c$;

**12**             Compute similarity vector $\mathbf{s}_c$ using Equation (1);

**13**         **end**

**14**         Compute the neighborhood vector $\mathbf{h} = [\mathbf{s}_1\|, \cdots, \|\mathbf{s}_C]$;

**15**         Compute the predicted vector with $\mathcal{F}$: $\mathbf{p} = \mathcal{F}(\mathbf{x})$;

**16**         Compute the trustworthiness $\mathbf{t}$ using Aggregator: $\mathbf{t} = \text{AGG}(\mathbf{W}_h\mathbf{h}, \mathbf{W}_p\mathbf{p})$;

**17**         Compute the loss function using Equation (4);

**18**         Update $\mathbf{W}_h$, $\mathbf{W}_p$, $\mathbf{W}$ via gradient descent;

**19**     **end**

**20** **end**

**21** **return** $\mathbf{W}_h$, $\mathbf{W}_p$, $\mathbf{W}$

---

backbone of the base classifier) are used for higher performance. Then, the final neighborhood vector $\mathbf{h}$ is constructed by concatenating all such class-wise similarity vectors:

$$\mathbf{h} = [\mathbf{s}_1 \| \mathbf{s}_2 \|, \cdots, \|\mathbf{s}_C], \tag{2}$$

where $\cdot \| \cdot$ denotes the column concatenation operator. The procedure is shown in the Figure 3.

**Aggregation.** Considering the potentially varying contribution of $\mathbf{h}$ and $\mathbf{p}$ to the trustworthiness score, we introduce two separate linear transformations to them, which are parameterized by $\mathbf{W}_h \in \mathbb{R}^{C \times CK}$ and $\mathbf{W}_p \in \mathbb{R}^{C \times C}$, respectively. We then aggregate the two resultant vectors using an operator $\text{AGG} : \mathbb{R}^C \times \mathbb{R}^C \mapsto \mathbb{R}^C$, which outputs a $C$-dimensional trustworthiness vector of $\mathbf{x}$ (one element for one class),

$$\mathbf{t} = \text{AGG}(\mathbf{W}_h\mathbf{h}, \mathbf{W}_p\mathbf{p}). \tag{3}$$

Here, the aggregation operator AGG can be instantiated by any neural network.

The optimization process is carried out by reducing the negative log-likelihood loss (NLL), i.e. $\mathbb{E}_{(\mathbf{x},\mathbf{y}) \sim (\mathcal{X}_{val}, \mathcal{Y}_{val})}[\mathcal{L}(\mathbf{x}, \mathbf{y})]$ where each sample's loss is calculated as

$$\mathcal{L}(\mathbf{x}, \mathbf{y}) = -\frac{1}{C} \sum_{c=1}^{C} \mathbf{y}_c \log(\mathbf{t}_c). \tag{4}$$

The overall training procedure is summarized in Algorithm 1.

For simplicity, a single-layer feedforward neural network with a learnable weight matrix $\mathbf{W} \in \mathbb{R}^{C \times 2C}$ and a nonlinear activation $\sigma(\cdot)$ is used as the aggregator in this paper. Formally, the trustworthiness vector $\mathbf{t}$ (as shown in Figure 3) can be expressed as

$$\mathbf{t} = \text{softmax}\left(\mathbf{W}^T\sigma\left([\mathbf{W}_h\mathbf{h} \| \mathbf{W}_p\mathbf{p}]\right)\right). \tag{5}$$

Underlying the learnable framework, how do these two pieces of information cooperate during the aggregation? Curious about this question, we also investigate the mechanism and show the empirical result in section 4.1.

---

**Algorithm 2:** NEIGHBORAGG-CMD

---

**Input:** Dataset $\{(\mathbf{x_i}, y_i, \hat{y}_i)\}_{i=1}^{N}$; Mislabeling rate $p$; Confidence level $\alpha$; Well-trained trustworthiness model NEIGHBORAGG.

**Output:** Mislabeled sample set $\mathcal{S}$.

1   $\mathcal{T} = \{\mathbf{t}_i \mid \mathbf{t}_i = \text{NEIGHBORAGG}(\mathbf{x}_i), \quad \forall\, 1 \le i \le n\}$

2   $\mathcal{R} = \{r_i \mid r_i = (2 \cdot \mathbb{I}(\hat{y}_i = y_i) - 1) \cdot t_{iy_i}, \quad \forall\, \mathbf{t}_i \in \mathcal{T}\}$

3   $\mathcal{R} = \text{sort}(\mathcal{R})$                                    ▷ *Sort in non-increasing order*

4   $B_\alpha = \lceil (N+1)(1-\alpha) + \alpha N p) \rceil$

5   $\tau_\alpha = r_{(B_\alpha)}$                                 ▷ $r_{(B_\alpha)}$ *is the $B_\alpha$-th largest element of $\mathcal{R}$*

6   $\mathcal{S} = \{(\mathbf{x}_i, y_i) \mid r_i \le \tau_\alpha, \quad \forall\, 1 \le i \le N\}$

7   **return** $\mathcal{S}$

---

**Inference.** Given a test sample $\tilde{\mathbf{x}}$, we construct its corresponding neighborhood vector $\tilde{\mathbf{h}}$ from the training dataset and fetch its classifier output vector $\tilde{\mathbf{p}}$ from the base classifier $\mathcal{F}$. Then we evaluate the trustworthiness vector $\tilde{\mathbf{t}}$ using equation 3 and fitted model parameters $\mathbf{W}, \mathbf{W}_h$ and $\mathbf{W}_p$. Finally, the trustworthiness score can be evaluated by indexing the trustworthiness vector using predicted class $c^*$, i.e., $\tilde{t}_{c^*}$.

### 3.2 Relation to Graph Neural Networks

In this section, we study the relations between our design and graph neural networks (GNNs) and show that our approach is inherently more flexible than GNNs in terms of aggregating neighborhood information to augment the classifier output for trustworthiness prediction. GNNs (Kipf & Welling, 2017; Xu et al., 2019) have been a topic of interest in recent times for their powerful modeling capacity to aggregate neighbors, and this motivates us to compare our method with GNNs. Among the several GNN variants, we choose the widely used graph convolutional neural network (GCN) as the subject for simplicity.

First, we show that our design of employing only one-hop neighbors for trustworthiness prediction is effective and efficient by comparing the performance of multi-hop GCNs with one-hop GCNs. Empirically, we demonstrate that the use of multi-hop GNNs does not have significant improvement and even degrades the performance for some datasets (see Table 2). We argue that multi-hop neighborhood aggregation may lead to the over-smoothing issue and the noise accumulation risk, at least for our task.

Second, we prove that NEIGHBORAGG is essentially a *generalized* version of a one-hop GCN: when imposing certain constraints on our NEIGHBORAGG (i.e., fixing the learned matrices $\mathbf{W}_h$ and $\mathbf{W}_p$ to be block diagonally-dominant), NEIGHBORAGG acts as a one-hop GCN. This equivalence is rigorously characterized as follows.

**Theorem 1** (One-hop GCN Equivalence). *Provided that $\mathbf{W}_h$ exhibits a block diagonal structure:*

$$\mathbf{W}_h = \frac{1}{K}\left[ I_{C \times C} \otimes \mathbf{1}^T \right] \text{ with } \mathbf{1}^T = \underbrace{[1, 1, \cdots, 1]}_{K\,1's},$$

*where $\otimes$ denotes the Kronecker product, and that $\mathbf{W}_p = I_{C \times C}$, NEIGHBORAGG operates as a one-hop Graph Convolutional Network with the node features $[\mathbf{y} \,\|\, \mathbf{0}] \in \Delta^{2C-1}$ for $y \in \mathcal{Y}_{tr}$ and $[\mathbf{0} \,\|\, \mathbf{p}] \in \Delta^{2C-1}$ for $\mathbf{p} = \mathcal{F}(\mathbf{x})$ with $\mathbf{x} \in \mathcal{X}_{val} \cup \mathcal{X}_{ts}$, and the adjacency matrix $\mathbf{A}$ induced by a predefined kernel $\mathcal{K}_f$ (e.g. Laplacian kernel).*

*Proof.* The proof is relegated to Appendix A.                                         □

**Remark.** In fact, our approach is more flexible than one-hop GCN for feature aggregation, as $\mathbf{W_h}$ and $\mathbf{W_p}$ in our setting can exhibit more flexible forms than the simple block diagonal structure. This is further verified by empirical studies (see Figure 6), which show that our model can exploit not only intra-class relations, but also inter-class relations, which one-hop GCN cannot. More detailed analyses can be found in Section 4.1.

### 3.3 Conformal Mislabel Detection: An Extension of NeighborAgg

In this section, we show that our trustworthiness score can also be used for another task, mislabel detection. In particular, we introduce the NEIGHBORAGG-CMD algorithm to assess the reliability of data labels in a mislabeled dataset. The detailed procedure is described in algorithm 2.

To identify mislabeled data in a noisy-labeled dataset, we compute a reliability score for the label of each sample using NEIGHBORAGG and a well-trained base classifier. The reasoning behind this is that *labels that contradict the classifier's prediction and have low trustworthiness scores are questionable*, meaning that when the neighborhood supports the classifier's prediction rather than the label itself, it is more likely the label that makes a mistake. So we devise the *reliability score* based on the class label $y$'s trustworthiness score $t_y$:

$$r = (2 \cdot \mathbb{I}(\hat{y} = y) - 1) \cdot t_y, \tag{6}$$

where the indicator function $\mathbb{I}(\cdot)$ and the classifier's prediction $\hat{y}$ are used to detect whether the sample is misclassified. Samples with reliability scores lower than a certain threshold $\tau_\alpha$ are treated as mislabeled (i.e. $r < \tau_\alpha$).

To bound the probability of false negative detections, we apply the conformal anomaly detection (Balasubramanian et al., 2014) framework and extend the existing work to the noisy setting for determining the threshold $\tau_\alpha$. Given a dataset of size $N$ with mislabeling rate $p$, and a user-specified confidence level $\alpha$, we compute reliability scores for each sample in the validation dataset and sort them in non-increasing order as $(r_{(1)}, \ldots, r_{(N)})$. The threshold $\tau_\alpha$ is set to the $B_\alpha$-th largest element, i.e.,

$$\tau_\alpha = r_{(B_\alpha)}, \text{ where } B_\alpha = \lceil (N+1)(1-\alpha) + \alpha N p \rceil. \tag{7}$$

Next, we show the following theoretical guarantee which to the best of our knowledge is the first to consider the real-world noisy data setting.

**Theorem 2** (Noisy-robust Coverage Guarantee). *For any given confidence level $\alpha \in (\frac{1}{N+1}, 1)$, with probability at least $1 - \alpha$ over the random choice of any correctly labeled data point $(\tilde{\mathbf{x}}, \tilde{y})$, we have*

$$\tilde{r} > \tau_\alpha,$$

*where $\tilde{r}$ is the predicted reliability score of $\tilde{\mathbf{x}}$ and $\tau_\alpha$ is defined in Equation (7).*

*Proof.* The detailed proof is relegated to Appendix B. In contrast to existing work in conformal learning which requires an i.i.d. validation set, our algorithm uses a partitioning approach to allow for a more realistic setting involving a small percentage of mislabeled samples. □

**Remark.** Theorem 2 suggests that the reliability score that our method outputs provide a theoretical guarantee — with high probability, a correctly labeled data point will be given a score above the threshold $\tau_\alpha$. In other words, if we select the likely mislabeled samples by selecting those below the reliability threshold, we can bound the probability of a false positive (by $\alpha$).

## 4 Experiments

Through extensive experiments, we aim to answer the following questions:

- **Mechanism Visualization:** How does NEIGHBORAGG work?
- **Effectiveness:** How well does NEIGHBORAGG perform on different types of datasets?
- **Ablation Study:** How does each component of NEIGHBORAGG contribute to the trustworthiness performance?
- **Sensitivity**: How sensitive to hyperparameters is NEIGHBORAGG?
- **Computational Cost**: How fast is NEIGHBORAGG?
- **Case Study:** How is NEIGHBORAGG extended to mislabel detection?

Due to space limitation, we refer discussions about hyperparameter sensitivity, computational cost, and case study of mislabel detection to the Appendix D, E, I.

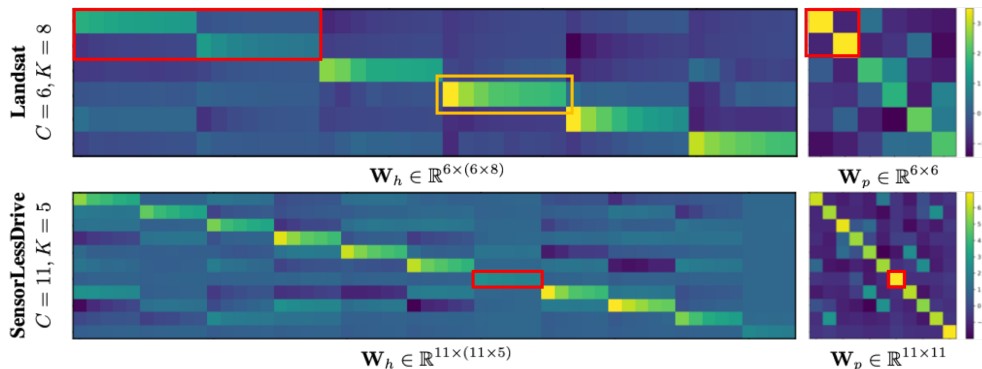

Figure 4: **Visualization of weight matrices $\mathbf{W}_h$ for similarity vectors and $\mathbf{W}_p$ for classifier output learned by NeighborAgg**. Brighter colors indicate larger values. The dimmer diagonal blocks in $\mathbf{W}_h$ (e.g. in the red rectangle) are empirically associated with their corresponding brighter diagonal entries in $\mathbf{W}_p$, suggesting that NeighborAgg combines sample confidence with neighborhood information in a *complementary* manner. The different weights within each diagonal block in $\mathbf{W}_h$ (e.g. in the orange rectangle) suggests that NeighborAgg can learn an appropriate $K$ (i.e. how many neighbors are necessary to determine the trustworthiness) for every class based on its local density, making it hyperparameter-insensitive.

## 4.1 Mechanism Visualization and Verification

In this section, we examine the mechanism of NeighborAgg empirically, demonstrating that our method utilizes neighborhood and classifier information in a complementary manner and captures intra-class and inter-class relations inside the neighborhood.

**Complementary effect**  We show that NeighborAgg integrates neighborhood information with classifier output in a *complementary* manner: it adaptively weighs the importance of the classifier's output against neighborhood information for each class. In other words, when the sample's neighborhood information is not accurate or useful, it relies more on the classifier output; and vice versa. To show this empirically, Figure 4 suggests that the dimmer diagonal blocks in $\mathbf{W}_h$ (e.g. in the red rectangle) are empirically associated with their corresponding brighter diagonal entries in $\mathbf{W}_p$, and vice versa. To further confirm this quantitatively, we calculate the Pearson's correlation coefficient $\rho$ between the diagonal blocks in $\mathbf{W}_h$ and the corresponding diagonal entries in $\mathbf{W}_p$ on Landsat. The result shows a strong negative correlation $\rho = -0.90$, which sheds light on the complementary mechanism in NeighborAgg.

**Intra-class and inter-class relations**  We have three observations on how NeighborAgg leverages neighbors: firstly, aligning with our motivation and Theorem 1, the learned weight matrices $\mathbf{W}_h$ and $\mathbf{W}_p$ are significantly block-diagonally dominant as shown in Figure 4, exhibiting the neighbor-homophily property and our method's similarity to graph neural networks. Secondly, the different weights within each diagonal block in $\mathbf{W}_h$ reflect different importance among neighbors of the same class (i.e. *intra-class* proximity) as shown in the orange rectangle region of Figure 4. This suggests that NeighborAgg can automatically learn how many neighbors are necessary to determine the trustworthiness for every class based on its neighborhood without tuning the hyperparameter $K$. Thirdly, the off-diagonal blocks represent the inter-class relations, such as similar or exclusive relations among classes, which makes it more flexible in determining a robust trustworthiness score. We leave the more detailed discussion in Appendix C.

## 4.2 Experiment Setup

**Datasets.**  We evaluate our method on image datasets including CIFAR10 (Krizhevsky, 2009), FashionM-NIST (Xiao et al., 2017) and MNIST (Deng, 2012), and UCI tabular datasets (Dua & Graff, 2017), including CardDefault, Landsat and LetterRecognition, etc. Statistics of each dataset are summarized in Appendix G.

| Clf | Method | LetterRecognition | | | Landsat | | | CardDefault | | |
|---|---|---|---|---|---|---|---|---|---|---|
| | | AUC % | APC % | APM % | AUC % | APC % | APM % | AUC % | APC % | APM % |
| LR | Confidence | 85.28(0.18) | 95.03(0.14) | 61.25(0.51) | 88.05(0.47) | 97.77(0.07) | 52.09(2.13) | 65.19(0.30) | 86.94(0.21) | 33.44(0.32) |
| | TempScaling | 84.67(0.22) | 94.83(0.15) | 59.72(0.64) | 87.20(0.42) | 97.63(0.08) | 48.85(1.73) | 65.22(0.33) | 87.04(0.25) | 33.50(0.36) |
| | TrustScore | 95.75(0.23) | 98.46(0.12) | 86.93(0.63) | 91.55(0.39) | 98.39(0.12) | 64.76(0.70) | 61.61(0.42) | 85.35(0.28) | 28.42(0.41) |
| | TCP | 90.78(0.21) | 96.96(0.13) | 74.85(0.43) | 89.47(0.49) | 98.06(0.15) | 54.25(1.78) | **68.79(0.29)** | **88.78(0.22)** | 34.14(0.37) |
| | TopLabel | 78.58(0.31) | 92.63(0.18) | 44.85(0.53) | 84.05(0.31) | 96.45(0.10) | 41.55(1.10) | 64.80(0.45) | 86.68(0.30) | 33.91(0.40) |
| | Ours | **99.08(0.04)** | **99.72(0.01)** | **97.17(0.13)** | **93.40(0.17)** | **98.84(0.04)** | **72.54(1.40)** | 67.60(0.31) | 87.45(0.22) | **36.06(0.33)** |
| RF | Confidence | 93.94(0.29) | 99.48(0.03) | 51.41(1.97) | 90.25(0.37) | 98.69(0.09) | 48.77(1.78) | 68.89(0.32) | 89.43(0.11) | 33.24(0.37) |
| | TempScaling | 94.58(0.19) | 99.55(0.02) | 55.41(1.86) | 89.26(0.24) | 98.56(0.05) | 46.88(0.91) | 68.68(0.32) | 89.31(0.16) | 33.07(0.44) |
| | TrustScore | 90.96(0.23) | 99.22(0.01) | 39.19(1.46) | 88.52(0.34) | 98.51(0.04) | 43.36(2.76) | 59.68(0.29) | 84.89(0.23) | 25.84(0.39) |
| | TCP | 85.83(0.22) | 98.71(0.06) | 29.40(0.53) | 85.07(0.77) | 97.97(0.16) | 34.30(1.62) | 67.96(0.12) | 89.57(0.14) | 30.08(0.24) |
| | TopLabel | 83.99(0.51) | 98.44(0.10) | 26.79(0.55) | 84.44(0.30) | 97.57(0.11) | 32.54(1.10) | 67.64(0.36) | 88.90(0.20) | 32.14(0.38) |
| | Ours | **96.45(0.18)** | **99.69(0.02)** | **72.16(1.36)** | **91.23(0.26)** | **98.91(0.06)** | **53.60(1.80)** | **69.27(0.30)** | **89.61(0.09)** | **34.27(0.45)** |
| MLP | Confidence | 90.71(0.18) | 99.18(0.04) | 39.59(0.93) | 84.41(1.61) | 96.95(0.71) | 40.26(2.04) | 68.99(0.32) | 89.05(0.18) | 34.17(0.49) |
| | TempScaling | 93.83(0.15) | 99.49(0.01) | 52.68(0.58) | 87.10(0.53) | 97.98(0.16) | 46.16(1.20) | 68.46(0.39) | 88.96(0.20) | 34.69(0.45) |
| | TrustScore | 88.53(0.31) | 99.05(0.05) | 32.28(0.60) | 88.09(0.48) | 98.32(0.09) | 41.85(1.47) | 60.20(0.39) | 84.82(0.25) | 26.60(0.35) |
| | TCP | 79.91(0.58) | 97.62(0.15) | 25.61(1.06) | 86.01(1.01) | 97.78(0.26) | 39.70(2.67) | 67.77(0.19) | 88.91(0.09) | 31.17(0.27) |
| | TopLabel | 78.78(1.55) | 98.09(0.20) | 16.65(0.79) | 81.29(1.02) | 96.70(0.29) | 30.16(0.93) | 67.64(0.37) | 88.68(0.15) | 32.81(0.44) |
| | Ours | **95.02(0.36)** | **99.58(0.04)** | **65.81(1.19)** | **91.75(0.39)** | **98.88(0.08)** | **57.80(0.91)** | **69.69(0.29)** | **89.51(0.19)** | **35.64(0.23)** |

Table 1: The performance of our proposed model NEIGHBORAGG and other models on three tabular datasets (**mean±std**). We report the results of all models based on different base classifiers (LR, RF, MLP) and best results are emphasized in bold. We use *TempScaling* for Temperature Scaling due to space limitation.

**Compared methods.** We compare our proposed NEIGHBORAGG with the following methods:

- **Confidence Score** (Hendrycks & Gimpel, 2017) employs the maximum softmax output of a classifier as a measure of trustworthiness.
- **Temperature Scaling** (Guo et al., 2017) modifies the confidence score using a temperature parameter $T$ learned from the validation set.
- **TCP** (Corbière et al., 2019) trains a ConfidNet using an intermediate output of any neural networks as the input for regression to the desired softmax output.
- **Trust Score** (Jiang et al., 2018) defines the trustworthiness measure as the ratio between the distance from the test sample to its nearest neighbor with labels excluding the predicted class, and the distance from the test sample to the nearest neighbor of the predicted class.
- **Top-label Calibration** (Gupta & Ramdas, 2021) calibrates a classifier's softmax output by using histogram binning to reduce top-label multi-class calibration into binary calibration.
- **GCN-khop** uses k-hop GCN to aggregate neighborhood information and classifier output for trustworthiness prediction. Its detailed implementation can be found in Theorem 1 and Appendix F.

**Evaluation Metrics.** Following the existing pioneering work on trustworthiness (Hendrycks & Gimpel, 2017; Corbière et al., 2019), we adopted the *same* metrics to evaluate the trustworthiness of a base classifier: AUC-ROC, APM and APC. More details regarding specific evaluation procedures can be found in Sec. 2 of Hendrycks & Gimpel (2017). All reported results are averaged over 5 trials under distinct random seeds on the same splits of datasets.

**Implementation Details.** For tabular datasets, experiments are conducted based on three base classifiers, including logistic regression (LR) (Peng et al., 2002), random forest (RF) (Svetnik et al., 2003) and multi-layer perceptrons (MLPs) (Ruck et al., 1990); while for image datasets, shallow convolutional networks, Resnet18 and Resnet50 (He et al., 2016) are used. We leave details such as hyperparameters to Appendix G.

### 4.3 Effectiveness of NeighborAgg

Performance results on tabular datasets and image datasets are summarized in Table 1 and Table 2 respectively, from which we make the following observations:

Firstly, our proposed NEIGHBORAGG outperforms other models under almost all metrics across benchmarks. Specifically, Table 1 shows that our model achieves the most significant improvement on APM, with the highest performance gain of 12.17%, and the average performance gain of 7.63%. This suggests that our model performs best in identifying misclassified samples. Besides, NEIGHBORAGG achieves more than 2%

| Method | MNIST | | | FashionMNIST | | | CIFAR10 | | |
|---|---|---|---|---|---|---|---|---|---|
| | AUC % | APC % | APM % | AUC % | APC % | APM % | AUC % | APC % | APM % |
| Confidence | 90.48(0.39) | 98.93(0.06) | 46.71(1.93) | 91.31(0.32) | 99.10(0.03) | 44.22(1.03) | 83.72(1.21) | 94.97(0.66) | 54.42(1.66) |
| TempScaling | 90.50(0.40) | 98.93(0.06) | 47.27(1.99) | 91.33(0.31) | **99.11(0.03)** | 44.27(0.99) | 83.75(1.29) | 94.96(0.68) | 54.81(1.80) |
| TrustScore | **96.40(0.28)** | **99.61(0.04)** | 78.53(1.25) | 91.31(0.21) | 99.10(0.03) | 47.27(0.86) | 86.98(0.75) | 96.04(0.18) | 63.29(3.84) |
| TCP | 92.11(1.03) | 98.37(0.43) | 69.91(7.47) | 90.82(0.07) | 98.83(0.03) | **50.52(1.33)** | 86.63(0.92) | 95.36(0.16) | 64.06(3.40) |
| TopLabel | 90.35(0.31) | 98.89(0.05) | 43.31(1.02) | 89.54(0.41) | 98.62(0.17) | 45.76(1.53) | 85.24(1.17) | 94.40(0.34) | 60.90(5.59) |
| Mahala | 75.88(0.23) | 96.91(0.03) | 21.96(0.48) | 59.87(0.95) | 94.26(0.06) | 11.15(0.08) | 55.73(5.92) | 82.93(3.07) | 23.47(3.10) |
| GCN3hop | 91.77(0.27) | 99.07(0.05) | 55.75(1.77) | 90.44(0.54) | 98.94(0.08) | 45.63(1.70) | 82.68(1.44) | 93.78(0.31) | 60.79(5.30) |
| GCN2hop | 91.58(0.29) | 99.05(0.05) | 54.71(1.90) | 90.35(0.56) | 98.92(0.07) | 45.63(1.91) | 83.75(1.66) | 94.18(0.41) | 62.37(5.46) |
| GCN1hop | 91.38(0.30) | 99.02(0.06) | 53.77(1.89) | 90.31(0.56) | 98.92(0.08) | 45.60(1.76) | 84.08(1.46) | 94.33(0.27) | 62.63(5.39) |
| Ours | **96.40(0.52)** | 99.55(0.08) | **81.02(1.91)** | **91.52(0.16)** | 99.04(0.02) | 48.83(0.51) | **87.52(1.07)** | **96.05(0.51)** | **65.27(4.03)** |

Table 2: The performance of NEIGHBORAGG and other models on image datasets (**mean ± std**). Best results are emphasized in bold. We refer to Table 1 for the full name of abbreviations.

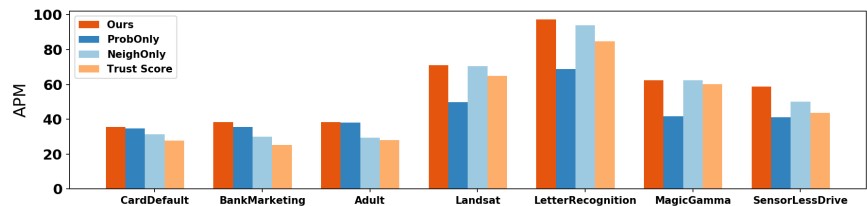

(a) Comparison for LR base classifiers under the APM metric.

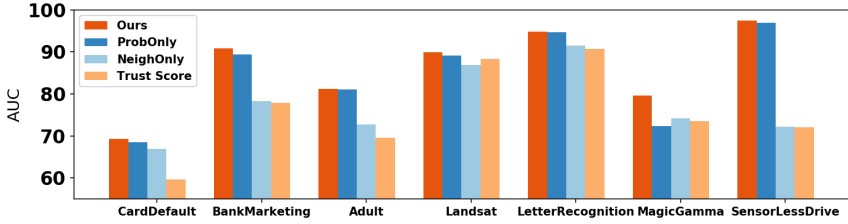

(b) Comparison for RF base classifiers under the AUC metric.

Figure 5: **Performance results for ablation study.** NEIGHBORAGG outperforms all the other model variants *ProbOnly* and *NeighOnly* across all datasets with LR and RF as the base classifiers, respectively. More results can be found in Appendix H.

improvement on AUC in most cases. Table 2 reveals that our model also achieves better or comparable performance on image datasets.

Secondly, the result suggests that the neighborhood information and classifier prediction are two essential and complementary sources of information for trustworthiness prediction. This is demonstrated by results shown in Table 1 that Trust Score achieves better results than Confidence Score on LetterRecognition and Landsat when LR is the base classifier, whereas Confidence Score performs better on CardDefault. Moreover, our method's outperformance of both information sources validates the complementary effect.

Thirdly, our method consistently beats the GCN-based method across all datasets, suggesting that our formulation is more effective and efficient. In addition, Table 2 also demonstrates that one-hop neighborhood aggregation is sufficient and that utilizing multi-hop neighbors may lead to the over-smoothing issue, by showing that multi-hop graph convolutional neural networks have limited improvement compared to the one-hop model, and sometimes become worse.

## 4.4 Ablation Study

To demonstrate the effectiveness of each component and the adaptiveness of the *learnable* weights in NEIGHBORAGG, we compare against its variants *ProbOnly* and *NeighOnly*,

| Detected mislabeled example questions |
|---|
| What is the past tense of past tense? |
| What is the difference between a fusion and a restaurant? |
| What are the new product for agent project? |
| Which protagonist from a video game have you most related to? |
| Do I have to appear for IMU CET again even if I get a good rank in it if I'm appearing for improvement of HSC board exam? |
| What are some important things/steps when starting a film production company in Netherlands? |
| What astrological combinations are needed to obtain a scholarship for studies? |
| What advice would you give a person intending to buy a Nissan note, in terms of performance i.e. traction, fuel economy, maintenance and resale? |

Table 3: Mislabeled samples detected in *QuoraInsQ* by our NEIGHBORAGG-CMD. These questions are labeled as insincere but are actually sincere.

- *NeighOnly*: solely takes neighborhood vectors as input,
- *ProbOnly*: solely inputs classifier output vectors,

as well as the non-learnable baseline Trust Score.

Figure 5 demonstrates the comparison results of the ablation study on seven tabular datasets. Results using other base classifiers and other metrics are listed in Appendix H.

We note that our NEIGHBORAGG consistently outperforms *ProbOnly* and *NeighOnly* across all datasets, especially on SensorLessDrive with 8.83% gain and on BankMarketing with 3% gain, which indicates that both vectors make non-negligible contributions to the final trustworthiness score, and that considering either of them alone is insufficient. It supports the claim that the neighborhood and the predictive output of the classifier complement one another in determining the trustworthiness score.

Moreover, the comparison between *NeighOnly* and Trust Score suggests that considering a set of neighborhood rather than solely class-wise nearest neighbors contributes to the performance gain; by inspecting those neighborhoods, NEIGHBORAGG can utilize richer information and flexibly choose its receptive field, i.e., how many neighbors are necessary to determine the trustworthiness score. It also empowers the model to adaptively capture intra-class relations (i.e. different proximity of a sample's neighbors) and inter-class relations, e.g., some classes may be more closely related as compared to other classes.

### 4.5 Mislabel Detection: A Case Study

This section demonstrates the usefulness of our NEIGHBORAGG-CMD algorithm by presenting mislabeled samples in real-world datasets. We use a dataset named *QuoraInsQ* from the Kaggle competition "Quora Insincere Questions Classification" that aims to improve online environment by detecting toxic questions. The dataset consists of 1,306,122 questions which are manually categorized as sincere or insincere. The definition of an insincere question is one that intends to make a statement instead of eliciting helpful responses. In order to estimate the mislabeling rate, we manually relabel 500 randomly selected questions and utilize the fraction of incorrectly labeled samples as the mislabeling rate.

Firstly, we use the model of the top-ranking team from the leaderboard as our base classifier and use NEIGHBORAGG-CMD to compute the reliability score for each sample. The mislabeling rate $p$ is estimated as 0.03. Then, we run NEIGHBORAGG-CMD with the confidence level $\alpha = 5\%$ and obtain the detected mislabeled results. Then we showcase some of the detected example questions with the lowest reliability scores in Table 3. We find that all of them were labeled as *'insincere'* in the original dataset, but none of them breach the four rules that signify a question as insincere. More experiments can be found in Appendix I.

## 5 Related Work

**Trustworthiness Prediction.** Trustworthiness prediction, also known as "failure prediction" and "misclassification detection" in the literature, aims to assign a discriminative score to every prediction given by a base classifier, indicating whether we can trust this prediction or not. This has received increasing

attention in recent times. Hendrycks & Gimpel (2017) suggests using confidence score to tackle this problem. Therefore, the confidence calibration method, designed to mitigate the over-confidence issue of the confidence score, can also be applied to trustworthiness prediction by giving a more accurate calibrated confidence score. Monte-Carlo dropout (Gal & Ghahramani, 2016) and Deep-Ensemble (Lakshminarayanan et al., 2017) compute the output variance of multiple trials to detect incorrect predictions, while these ensemble-based methods are quite computationally expensive. Trust Score (Jiang et al., 2018) proposes a score which is a fixed, non-learnable function of the neighborhood of a sample, and hence suffers from limited functional space. Corbière et al. (2019) proposes a regression method to fit the ground truth label's corresponding softmax score and uses it as a proxy for trustworthiness. However, this relies on the assumption that the base classifier is always over-confident, which is not always the case (e.g. graph neural networks were found to be under-confident in Wang et al. (2021)). Malinin & Gales (2018); Malinin et al. (2020); Sensoy et al. (2018); Charpentier et al. (2020) assume the classifier outputs are sampled from a latent Dirichlet distribution and treat low-likelihood samples as misclassified samples. These methods typically involve modified architectures that need to be trained from scratch, and in some cases can involve the trade-off between classification accuracy and the performance of trustworthiness prediction. In contrast, our proposed NEIGHBORAGG keeps the base classifier intact and uses auxiliary information for simple estimation. In our work, we aim to measure trustworthiness by adaptively utilizing the classifier's predictive output and neighborhood information via a flexible mapping that combines the best of both worlds.

**Relations to Out-of-distribution Detection.** Out-of-distribution (OoD) detection (Hendrycks & Gimpel, 2017; Sastry & Oore, 2020; Ming et al., 2022) are closely related to trustworthiness prediction but targeted at a different goal. OoD detection aims to measure the sample quality by identifying input data whose ground truth label is not covered by the label set of the training dataset, whereas trustworthiness prediction aims to measure the classifier's quality by identifying input data whose predicted label does not match its ground truth label. However, since they both detect abnormal behaviors, the assumption used for one task can be applied to the other with some modification. For example, our neighbor-homophily assumption can also be extended to OoD detection by assuming that samples that are distant from neighbors in all classes are likely to be out-of-distribution data. On the other hand, methods for detecting OoD data, such as mahalanobis distance Lee et al. (2018), can also be adapted to our task.

**Mislabel Detection** The goal of mislabel detection is to identify data whose labelled class differs from the underlying ground truth class. A majority of research in this field leverages training dynamics for differentiating correctly labelled and mislabelled samples, such as the dynamics of logit in AUM Pleiss et al. (2020) and the loss distribution in DY-BootstrapeArazo et al. (2019). Our approach falls into a different line of direction Northcutt et al. (2019); Zhang et al. (2021) that employs pre-trained classifiers. Confident Learning Northcutt et al. (2019), for example, estimates a noise transition matrix based on the softmax output. In addition to the softmax output, we also consider a sample's local neighborhood, which allows us to infer individual samples more accurately compared to confident learning.

## 6  Conclusions and Discussion

**Conclusion** Knowing when to trust a classifier is essential for safe deployment of present machine learning algorithms. To solve the problem, we devise a model-agnostic post-hoc trustworthiness prediction algorithm NEIGHBORAGG which leverages information from the neighborhood and the classifier to predict the trustworthiness of predictions given by a classifier. By theoretically demonstrating that NEIGHBORAGG is a generalized one-hop GCN aggregating information from the neighborhood and the sample itself, we provide a better understanding of how our approach works. The working mechanism is also revealed by empirical studies, which show that our approach can utilize the classifier output and neighborhood information in a complementary manner, and capture diverse similarities within different neighborhoods. On several tabular and image benchmarks, the effectiveness of NEIGHBORAGG is empirically validated via comparison with other state-of-the-art methods and ablation studies. Of independent interest, an extension of NEIGHBORAGG to mislabel detection is also introduced with a noise-robust coverage guarantee for bounding the false negative predictions.

**Discussion**   The current study assesses a classifier's trustworthiness by utilizing each sample's neighborhood, comparing each sample's prediction to its neighbors' ground truth labels. Besides that, it would be interesting to explore alternative information associated with the trustworthiness of the classifier. The model explanation generated by explainable approaches, for example, can be used to determine trustworthiness. A prediction with implausible explaining logic is likely wrong. Moreover, going beyond unstructured data, we believe NeighborAgg is also promising for measuring trustworthiness of graph-structured data, which can be an interesting and nontrivial extension to this work.

### Acknowledgments

This work was supported in part by NUS ODPRT Grant R252-000-A81-133.

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
