# OpenReview forum: "Birds of a Feather Trust Together: Knowing When to Trust a Classifier via Adaptive Neighborhood Aggregation"
_TMLR — Accepted by TMLR_

### Review · Reviewer_QWSV · 2022-06-28

**Summary Of Contributions:**

The paper studies the effect of incorporating class-wise k-nearest neighbor (kNN) models (in the input space) to calibrate the prediction confidences from a given classifier, to address the “trustworthiness prediction” task - a narrower variant of confidence calibration or out-of-distribution (OOD) detection tasks, where the focus is to define a score function capable to discriminate whether a given (test) input will be correctly/incorrectly classified. The paper next presents that the proposed method can be extended for detecting mislabeled samples. Experimental results on tabular (LetterRecognition, Landsat, CardDefault) and image (MNIST, FashionMNIST, CIFAR10) datasets show that the proposed method (score) achieves better AUC and Precisions on detecting misclassified samples over other baseline considered.

**Broader Impact Concerns:**

It seems the paper currently does not contain discussions that explicitly or implicitly states on Broader Impact.

**Requested Changes:**

Major requests

- More empirical investigation on the scalability of the proposed method, with respect to (a) the dataset size, (b) model size, and (c) # classes, as mentioned above. For example, I suppose that applying the method to pre-trained ImageNet classifiers would not that be technically hard given the model-agnostic sense of the method. If it is not the case, please add a respective discussion on why extending the method to larger-scale is practically difficult.
- More discussions and comparisons considering recent literatures of confidence calibration or OOD detection in the experiments. The Mahalanobis [1] baseline seems crucial in this point, given that it defines a new confidence score $p(y|x)$, and is a post-hoc algorithm from a pre-trained model, as also claimed in the paper.
- The definition of Trustworthiness prediction task (Problem 1) is quite unclear to me: Given that one gets trustworthiness scores $t_c$, what would be the goal of the task? - a more precise definition of the problem seems to be needed.

Minor requests

- Table 2: Missing indication in the main text for the details on how the GCN-based baselines (GCN*hop) are designed.
- Section 3.2: I feel that the relationship between the method to GCN is somewhat vague, given that the proposed (MLP-like) architecture is quite general in practice - a more discussion needed on the actual implication of this relationship would be appreciated, e.g., does it mean that the method can leverage some useful properties of GCN? Or, does NeighborAgg can improve GCN?
- There is also a literature on the mislabel detection task, e.g., [7], thus I think there should be also a respective discussion on it.

[7] Pleiss et al., Identifying Mislabeled Data using the Area Under the Margin Ranking, NeurIPS 2020.

**Strengths And Weaknesses:**

Strengths

- The paper is overall clearly written - it contains a clear motivation, a well-explained method, and detailed experimental setups (in Appendix).
- The experiments are thorough especially in its ablation study: i.e., to verify that having both information of prediction probability and neighborhoods complementarily improves confidence calibration.
- The proposed method is generally applicable for any pre-trained classifiers.

Weaknesses

- I think the paper could compare (or discuss) with more recent methods in the literature of confidence calibration or OOD detection in their experiments: in my understanding, the paper tackles the problem of “trustworthiness prediction” which is highly related to confidence calibration or OOD detection. The paper differentiates its position compared to those tasks in Section 5, but I generally feel that the differences are only weakly justified:
    - (a) It first states that the usual confidence calibration techniques are not suitable for trustworthiness prediction because they do not alter the ranking of the prediction: but I do not think it is a crucial issue, e.g., one can still define the max-confidence as the trustworthiness score to address the task, and I am not fully convince that these two tasks are essentially different.
    - (b) Compared to OOD detection, on the other hand, the paper states that the goal of OOD detection is not to evaluate on $p(y|x)$ but rather on $p(x)$: but this is also not clear to me as well - for example, one of state-of-the-art OOD detection method of Mahalanobis distance-based detection [Lee et al., 2019] does define $p(y|x)$ by means of a generative classifier based on a class-conditional Gaussian model, so one can still use this score to define a new trustworthiness score. Even for the method proposed in this paper, one may view that the method incorporates another generative classifier based on a kNN based model, so the method is related to Mahalanobis in this sense. The readers may also question whether other recent OOD based methods indeed only target modeling $p(x)$ rather than modeling a better $p(y|x)$ [2-6].
- The empirical gains are often not clear given the standard deviations, especially for the results on image datasets (Table 2). For the tabular results in Table 1, on the other hand, I wonder if the gains would still remain significant if one considers a larger base classifier, e.g., beyond an MLP, so that the # parameters of the classifier can dominate those newly introduced in the proposed method.
- The experiments are limited to only small-scale datasets, e.g., the 10-way CIFAR-10 classification at best, despite that one of the main claim of the proposed method is in its model-agnostic sense. There are many dimensions the paper did not cover to verify the scalability: e.g., with respect to the dataset size, model size, and # classes.

[1] Lee et al., A Simple Unified Framework for Detecting Out-of-Distribution Samples and Adversarial Attacks, NeurIPS 2018.

[2] Lee et al., Training Confidence-calibrated Classifiers for Detecting Out-of-Distribution Samples, ICLR 2018.

[3] Hendrycks et al., Using Self-Supervised Learning Can Improve Model Robustness and Uncertainty, NeurIPS 2019.

[4] Bergman and Hoshen, Classification-Based Anomaly Detection for General Data, ICLR 2020.

[5] Tack et al., CSI: Novelty Detection via Contrastive Learning on Distributionally Shifted Instances, NeurIPS 2020.

[6] Sastry et al., Detecting Out-of-Distribution Examples with Gram Matrices, ICML 2020.

---

> ### Author Response · Authors · 2022-07-12
> **Response to Reviewer QWSV (1)**
>
> **More empirical investigation on the scalability of the proposed method**:
> Thanks for the valuable suggestion. Regarding the scalability with respect to the dataset size, model size and \# of classes, we first empirically investigate the specific elapsed time on CIFAR10 and then analytically study how it scales to large datasets such as ImageNet. We argue that computational cost is unlikely to become the bottleneck for the deployment of our approach by proving that our method scales linearly with the number of neighbors $K$ and classes $C$, and approximately logarithmically with the number of samples $N$. Furthermore, a variety of acceleration techniques and scalable algorithms are available to further enhance scalability. For more details about the analysis, we kindly refer you to Appendix F or our first comment to reviewer tWn7 (https://openreview.net/forum?id=p5V8P2J61u&noteId=5VLYPLCtTX).
>
> **More discussions and comparisons considering recent literature of confidence calibration or OOD detection in the experiments**:
> We appreciate your insightful discussion and suggestion. It is true that calibration methods are able to perform the task of trustworthiness prediction. In fact, we have already included two calibration methods in our experiment setting: the most popular method, temperature scaling, and a more recent method, top label calibration. Moreover, we have also revised the related works in the main text (Sec. 5) to incorporate calibration methods as a line of research for trustworthiness prediction.
>
> Regarding the relation between OOD detection and trustworthiness prediction, we apologize for any confusion caused. When we mention that OOD Detection focuses on $p(x)$, we do not mean to say that the label is not considered or modeled. Instead, many OOD detection algorithms leverage $p(y|x)$ or $p(x|y)$. For example, the Mahalanobis distance-based algorithm can be seen to models $p(x)$ as $p(x) = \sum_{c=0}^C p(x|y=c)P(y=c) = \sum_{c=0}^C p(x|y=c)$ by implicitly assuming the label class follows a uniform distribution. Intuitively, if a sample has low $p(x|y=c)$ across all classes $c \in \\{1, 2, ... C\\}$, they should have a low $p(x)$. Therefore, the mahalanobis distance-based score is set to be $\max_c \log p(x|y=c) $ empirically.
>
> As you discussed, this modelling way also allows us to utilize $p(y|x) = \frac{p(x|y)p(y)}{p(x)} \propto p(x|y)$ for trustworthiness prediction. Therefore, we also include a modified Mahalanobis distance-based confidence score as one of our compared methods. The experiment results are updated to Table 2 of the main text. However, the results of Mahalanobis distance-based confidence score are not as promising as we expected, indicating that more consideration is needed when applying OOD detection algorithms to trustworthiness prediction.
>
> **A more precise definition of problem 1**: Thanks for proposing this advice. We have revised the problem 1 statement in Sec. 2 and added a formal description to describe the goal of this task. In short, the trustworthiness score for a correctly classified sample should be close to 1 and that for a misclassified sample should be close to 0 in perfect condition.
>
> **Table 2: Indication in the main text for the details on how the GCN-based baselines (GCN*hop) are designed**: Thanks for the valuable suggestion. We have added the GCN-based baseline to Sec 4.2 `Compared methods` and more implementation details to the Appendix G `Graph Neural Network Framework`(changes are highlighted in blue).
>
> In summary, the graph is constructed by setting edges between every sample and its $K$ nearest neighbors. Let us create a $C$-dimensional zero vector as a dummy vector. For the training data points, the node feature is constructed as the label's one-hot embedding concatenated with a dummy vector; while for the validation and test dataset, the node feature is constructed as a dummy vector concatenated with the softmax output of the sample. The edge feature is set as the similarity between the two samples.

---

> ### Author Response · Authors · 2022-07-12
> **Response to Reviewer QWSV (2)**
>
> **Discussion on the actual implication of this relationship**: Thank you for the valuable feedback.
> - Firstly, Theorem 1 shows that when $W_1$ and $W_2$ are fixed, our method acts as a one-hop GCN. This implies that our method is processing and combining information from its neighbors and its own features as the graph convolutional neural network does, by simply feeding neighbors' similarities rather than constructing a graph explicitly.
> - Secondly, this provides us with a tool for investigating some properties of our framework. For example, by utilizing these features, we can explore what kind of neighborhood is more suitable for trustworthiness prediction. By doing experiments on one-hop GCN and multi-hop GCN, for instance, we can fairly say that utilizing one-hop neighbors is sufficient compared to using multiple hops of neighbors, which is also more efficient.
> - In addition, we demonstrate that our method actually overcomes the underlying restriction of GCN and achieves better flexibility, by showing that we do not impose any constraints on $W_1, W_2$.
>
> **A respective discussion on the literature of the mislabel detection task**: Thanks for the helpful suggestion. We have added the discussion on the literature of the mislabel detection task to Sec. 5 (highlighted in blue color). For your convenience, we give a compact summary here.
>
> Studies on mislabel detection can be roughly categorized as training-dynamics-based and pretrained-model-based. Training-dynamics-based methods need to train a classifier from scratch on the targeted data. For instance, the AUM paper you mentioned [1] leverages the difference of the change tendency between the label class's logit value and the other largest logit value across every epoch, which requires training on these data from scratch. Our method, which leverages the difference in local neighborhood and classifier output, falls into the category of pretrained-model-based approaches.
>
> **Performance gain in terms of tabular data and image data**:
> - Firstly, we would like to clarify the reason why we choose MLP as the "large" base classifier for tabular datasets. On one hand, we find MLP is already sufficient in terms of classification for the tabular dataset. On the other hand, \# of parameters can be enlarged by increasing the \# of layers.
> - Secondly, Figures 1c and 1d in the main text show that the \# of parameters of the classifier will not erase the performance gains of the proposed method. In this experiment, we use a relatively large classifier (12 layers MLP) for 2-dimensional data and achieve 100\% accuracy on the training dataset. But this seemingly large classifier's information does not dominate the newly introduced neighborhood information; instead, the neighborhood information is utilized by our method to balance the effect of the overfitted classifier to give a more accurate trustworthiness score.

---

### Review · Reviewer_D4Gw · 2022-06-28

**Summary Of Contributions:**

This paper aims to obtain a trustworthiness score that can detect incorrect predictions of a pre-trained classifier. To this end, the authors propose to train a failure predictor whose inputs are the sample's neighborhood information and the classifier's output. The failure predictor is flexible enough to utilize the patterns in the inputs, thus achieving state-of-the-art trustworthiness performance.

**Broader Impact Concerns:**

None.

**Requested Changes:**

- (major) The statement of Problem 1 is not informative. The statement only tells the reader what information is required to calculate the trustworthiness score, while it does not define the goal of the score. What problem is the trustworthiness score proposed to solve? After reading the current statement of Problem 1, readers may not understand what Trustworthiness Prediction is.
- (major) From Algorithm 1, the failure predictor NeighborAgg is trained using a training set. Is the base classifier 100% accurate on this training set? It is a common case for deep neural networks to achieve 100% accuracy on a training set. If it is the case, then there is no need to detect an incorrect prediction on the training set. How can the proposed approach really learn to predict failure in this case?
- (minor) The inputs of Algorithm 1 include unused items, such as Validation set and Trustworthiness performance evaluation metric. These items do not appear in the the algorithm.
- (minor) Random forests are used in experiments. It would be better to explain what is the classifier output feature of a random forest for feature construction.

**Strengths And Weaknesses:**

**Strengths**
- This paper constructs reasonable features that represent the sample's neighborhood information.
- This paper demonstrates that the sample's neighborhood and the classifier's output are complementary in failure prediction.
- The proposed approach has good scalability: more information, such as model explanation scores, might be aggregated to obtain better performance.
- The empirical performance of the proposed approach is good, especially on tabular datasets.

**Weaknesses**
- Previous work has shown that the neighborhood information can be leveraged to measure the trustworthiness of a classifier's predictions. This work aims to show that there is still some room for performance improvement with this information. Then, the authors replace the non-trainable function of the previous work with a trainable one. In this sense, the proposed approach is marginal.
- Only two factors are leveraged to predict trustworthiness. More factors would be helpful.

---

> ### Author Response · Authors · 2022-07-07
> **Response to Reviewer D4Gw**
>
> Thank you for the valuable comments and feedback. We will address your concerns point by point as below:
>
> **[Novelty regarding the proposed method]** It is true that Trust score has already shown that neighborhood information can be used for trustworthiness prediction. However, it relies solely on neighborhood information, which could fail if the neighborhood is not computed accurately or the neighborhood-homophily assumption does not hold perfectly. In this sense, we are not just simply replacing the non-trainable function of the previous work with a trainable one, we are also opening the door to allow more informative factors, such as the logit scores of the base classifier we discussed, to be considered by providing a framework for aggregating different information. Furthermore, if additional suitable factors emerge that are not covered in this paper, the users can also easily incorporate them into the aggregation loop for trustworthiness prediction by following this framework.
>
> **[The statement of Problem 1 is not informative.]** Thanks for the suggestion. We have revised the problem 1 statement in Sec. 2 and added a formal description to describe the goal of this task. In short, the trustworthiness score for a correctly classified sample should be close to 1 and that for a misclassified sample should be close to 0 in perfect condition.
>
> **[Issue regarding Algorithm 1]** Apologies for the confusion caused. I have modified Algorithm 1 in the main text. In summary, we clarify that we use the training dataset to compute the nearest neighbors while using the validation set to train our proposed model, the aggregator. The test dataset is not used during training. In this case, even if the base classifier is 100\% accurate in the training dataset, it may not achieve 100\% accuracy in the validation, therefore, there are always failure modes they can learn. For instance, the base classifier in Figure 1c is 100\% accurate in the training dataset, with obvious overfitting. But by introducing the validation set, our model learns a more smooth decision boundary and mitigates the issue.
>
> **[Issue regarding training/validation/test dataset and unsued items]** Apologies for the confusion caused. I have revised Sec 3.1 and Algorithm 1 to make it clearer and highlighted changes in blue. In summary, we use the training dataset to compute the nearest neighbors while using the validation set to train our proposed model, NeighborAgg. The test dataset is not used during training.
>
> **[Explain what is the classifier output feature of a random forest]** For the random forest algorithm, the classifier output feature of an input sample is computed as the mean predicted class probabilities of the trees in the forest; the class probability of a single tree is the fraction of samples of the same class in a leaf. During implementation, we call the random forest function from the scikit-learn library and use the predict\_proba function in the random forest class for deriving the classifier output vector. Details can be found in this link: https://scikit-learn.org/stable/modules/generated/sklearn.ensemble.RandomForestClassifier.html.

---

### Review · Reviewer_tWn7 · 2022-06-29

**Summary Of Contributions:**

The paper proposes a technique for estimating the trustworthiness of the predictions of an ML model. The proposed technique uses the logits of the model, together with neighborhood information. The method also extends to a technique for detecting mislabeled data instances.

**Broader Impact Concerns:**

None that I can think of.

**Requested Changes:**

I would suggest the following changes, which will hopefully make the contributions of the paper more clear and improve readability:

Major:
- Please add a discussion on computational and memory aspects of the proposed methods and compare to prior work.
- Please add a discussion on Theorem 2, in particular how does it depend on $\mathcal{F}$.
- Overall, it will be nice to make it more clear what sections/paragraphs/equations/part of the algorithm work with the training, validation and test data.

Simple, but important fixes:
- Page 3, Problem 1: Does the trustworthiness score depend on the test labels as well? This is currently imply by $t_c = \Tau (\mathcal{F}(x); \mathcal{F}, \{\mathcal{X}, \mathcal{Y}\})$.
- Page 3: $\mathcal{X}$ is used both for the input space and for the set of inputs in the dataset.
- Page 4: what is the function $f$ used for the similarity measure?
- Page 5, before equation (4): do you mean expectation w.r.t. $\mathcal{X}_{tr}$, not $\mathcal{X}_{val}$
- Page 5, Inference: how is the neighbourhood vector $\tilde{h}$ computed? Is it with respect to the training data points? With respect to the other test points?
- Algorithm 2 and Section 3.3 - please use indexes for the data points, otherwise the notation is very confusing.
- Algorithm 2: what is $n$? Do you mean $N$?

Minor:
- Page 2: by focusing on a pretrained classifier, and among which ... --> by focusing on a pretrained classifier. Among these ...
- Page 3 footnote: dataset --> datasetS
- Page 4, Feature reconstruction: make it clear that this is about the training data - at test time you do not have access to the labels.

**Strengths And Weaknesses:**

Pros:

- The paper addresses an important problem, namely accessing the reliability of ML predictions.
- The proposed method is intuitive and relatively clearly explained (see questions below though), and it seems to perform reasonably well in experiments.
- Comparison to related work is nice.
- The paper is relatively well-written.

Weakness:

- Some aspects about the algorithm are not very clear to me, see "Requested changes".
- The method seems to incur a high computational and memory cost at inference time. This does not seem to be discussed in the paper.
- Theorem 2 is fairly confusing to me, since it does not seem to depend on the underlying classifier $\mathcal{F}$ at all. Is it purely a probabilistic statement related to the random sampling of the data and the properties of ordering?

---

> ### Author Response · Authors · 2022-07-07
> **Response to Reviewer tWn7 (1)**
>
> Thank you for the valuable comments and feedback. We will address your concerns point by point as below:
>
> **[Discussion on computational cost at inference time]**:
> We have a discussion regarding the computational cost, which are put in the appendix due to space limitation. For your convenience, we have included the discussion here and updated the modified version to the Appendix F.
>
> **Time complexity** We start by giving an intuitive example of the specific elapsed time of our method and then analyze how our method scales with \# of classes, dataset size and model size.
>
> Empirically, taking CIFAR10 dataset as an example, we provide the inference time of our method for 10000 samples on 1 Tesla V100 SXM2 GPU with 32GB memory and Intel Xeon CPU E5-2698 v4 @ 2.20GHz with 504GB available memory. The neighbor search is executed on either GPU (faiss) or CPU (scikit-learn). To make the time cost more intuitive, we also provide the elapsed time of ResNet18 classifying 10000 samples as a baseline comparison. The results shows that GPU-accelerated neighbor search can be 3 times faster than the CPU version, and the CPU version spends comparable time with a common classification network.
>
> |               | GPU (min) | CPU (min) | Baseline |
> |---------------|-----------|-----------|----------|
> | 10000 queries | 0.238     | 0.768     | 0.790    |
>
> Analytically, the inference time complexity of our proposed method is determined by two components: neighbor search and aggregation, with the former shared solely by methods requiring neighborhood information, such as Trust Score and ours.
> - First, the aggregation operator is instantiated as a neural network, whose time complexity is determined by the number of layers of the network. This is similar to most methods in this field, such as TCP, suggesting that our method is similar to the others in the worst case. However, our aggregator is usually faster since it has fewer layers (e.g., one layer in our implementation), which is because it only deals with low-dimensional auxiliary information (e.g., similarities to neighbors) rather than high-dimensional feature embedding. In this sense, the aggregation process does not incur high computational cost.
> - Second, without placing a high priority on computational cost, we have chosen the simple and widely-used kd-trees from scikit-learn library for nearest neighbor search. It is worth mentioning that neighbor search is a well-studied problem for which numerous algorithms have been developed with different benefits. In terms of time complexity, there are many algorithms aiming to reduce the computing time, such as locality-sensitive hashing and randomized kd-trees, etc[1].
>
>   For our implementation, the average time complexity of the neighbor search phase is $O(Kd \cdot \sum_{c=1}^C \log N_c)$ where $K, C, d$ are the \# of neighbors, classes and embedding dimension, and $N_c$ denotes the size of the kd-tree for class $c$ (the \# of samples for class $c$ in the training dataset).
>   - First, a small value of $K$ is sufficient for most cases (e.g., $K = 20$ in our implementation) and it does not increase too much with $N$. Besides, because the kd-tree organizes similar samples together, after a single query we can simply search the surroundings around that point to find other nearest neighbors. Therefore, the implementation can be optimized by querying only once [2].
>   - Secondly, the size of dataset $N$, has limited impact due to the logarithmic scaling. For example, as $N$ increases from $0.05$M (CIFAR10) to $1.20$M (ImageNet), the query time only increases by around 1.3 times, which is fairly insignificant.
>   - Thirdly, even though it grows linearly with $C$, each class's associated kd-tree can be executed in parallel to mitigate the effect, especially for tasks with a large value of $C$, such as $C=1,000$ in ImageNet, since they are independent of each other. Besides, the computational power of GPU and other hardware acceleration techniques can also be utilized, as demonstrated above.
> - Regarding model size, it does not affect scalability because the input of our technique is the classifier's output, which has been fixed to dimension $C$.
>
> Therefore, we argue that computational cost is unlikely to become the bottleneck for the deployment of our approach.
>
> [1] Andoni, Alexandr, and Piotr Indyk. "Nearest neighbors in high-dimensional spaces." Handbook of Discrete and Computational Geometry. Chapman and Hall/CRC, 2017. 1135-1155.
>
> [2] https://www.colorado.edu/amath/sites/default/files/attached-files/k-d_trees_and_knn_searches.pdf

---

> ### Author Response · Authors · 2022-07-07
> **Response to Reviewer tWn7 (2)**
>
> **[Discussion on memory cost at inference time]** The memory cost of our method can also be decomposed into neighbor search and aggregation cost. As we have illustrated before, the aggregator typically has fewer layers and smaller input dimensions, hence incurring little memory cost. As for neighbor search, the memory cost is dependent on the size of all kd-trees $N$ and the feature dimension $d$, with an analytical space complexity of $O(Nd)$. This analysis applies to all methods requiring neighbor search, including Trust Score. Typically, the feature dimension $d$ for complex data, such as images, is approximately proportional to dimension of manifold, which is substantially lower than the original data dimension. For example, ImageNet with feature dimension $d=256$ consumes less than $300$MB of memory, which can be deployed on CPU or GPU. Moreover, memory-efficient kd-tree~[1-2] and training dataset compression techniques can also be used for reducing memory cost further.
>
> **[Discussion on Theorem 2, in particular how does Theorem 2 depend on $\mathcal{F}$.]** Apologies for the confusion caused. Before explaining Theorem 2, we would like to clarify the motivation of why we propose it. Whether it is for mislabel detection or failure prediction, if we want a binary decision for determining whether the prediction on the sample is correct or not, we need a threshold to transform the original output to binary output. Then the question arises: how to select a good threshold?
> Inspired by the conformal anomaly detection framework, we want to set a confidence level $\alpha$ for false negative rate and compute a reasonable threshold based on $\alpha$. Therefore, Theorem 2 states that if we want the false negative rate to be less than $\alpha$, we should choose the threshold as stated in equation (7).  The proof of this theorem is mainly based on the i.i.d. setting of clean data in the test and validation sets. Intuitively, if 95\% clean samples ($\alpha=0.05$) have reliability scores larger than the threshold, it is also fair to assume that 95\% of the test samples should have a reliability score greater than the threshold, given that they come from the same distribution. In other words, a sample with a score greater than the threshold has a 95\% chance of being correctly labeled. In order to apply it to a noisy setting, we made some modifications but the intuition remains the same.
>
> Regarding how Theorem 2 depends on the base classifier, the base classifier influences the selection of the threshold by affecting the value of each sample's reliability score and, consequently, the ranking of each sample's reliability score, which decides the value of the threshold.
>
> **[Issues about training/validation/test dataset; before equation (4): do you mean expectation w.r.t. $ X_{tr} \text{ or } X_{val}$]** Apologies for the confusion caused. I have revised Sec 3.1 and Algorithm 1 to make it clearer and highlighted changes in blue. In summary, we use the training dataset to compute the nearest neighbors while using the validation set to train our proposed model, NeighborAgg. The test dataset is not used during training.
>
> **[Does the trustworthiness score depend on the test labels as well?]**
> No. The trustworthiness score is used to measure the quality of a prediction class, so it should depend on the predicted class label $c = \hat{y}$. However, it does not depend on the ground truth label.
>
> **[What is the function $f$ used for the similarity measure?]**
> For tabular dataset, the transform $f$ of the similarity kernel $\mathcal{K}_{f}$ is set to be identity mapping of the original data. Empirically, we find that learning with identity mapping yields sufficiently good performance. A more complex transform (e.g. a pretrained network) can be used to achieve higher performance. For image datasets, the function $f$ is constructed using the backbone feature extractor of the base classifier. We have added this to Sec 3.1.
>
> **[Algorithm 2: what is n? Do you mean N?]**
> Yes, it means N. We have modified this typo in the original paper.
>
>
> [1] Choi, Byeongjun, Byungjoon Chang, and Insung Ihm. "Improving Memory Space Efficiency of Kd‐tree for Real‐time Ray Tracing." Computer Graphics Forum. 2013.
>
> [2] Rafiee, M., and M. Abbasi. "Pruned Kd-tree: a memory-efficient algorithm for multi-field packet classification." SN Applied Sciences 1.12 (2019): 1-19.

---

### Author Response · Authors · 2022-07-12
**To All Reviewers**

We thank all reviewers for the constructive comments and valuable suggestions. In the revised paper, we highlighted the main changes in blue. Based on each reviewer's concerns, we summarize the main changes of our paper below:
- We have revised the statement of Problem 1 in Sec. 2 and added a formal description for the goal of this task.
- We have revised Sec 3.1 and Algorithm 1 to make the use of the training/validation/test dataset more explicit.
- We have added the description of GCN-based baseline to the Compared methods part of Sec 4.2 and more implementation details to Appendix G (Graph Neural Network Framework).
- We have added the discussion on the literature of the mislabel detection task to Sec 5 Related Works.
- We have revised the relations to Out-of-distribution detection to make it more explicit.
- we have revised the related works in the main text (Sec 5) to incorporate calibration methods as a line of research for trustworthiness prediction.
- We have modified Appendix F (Computational and Memory Cost) to analyze the scalability of our proposed method.

---

### Decision · Action_Editors · 2022-07-31

**Recommendation:** Accept with minor revision

**Comment:**

The paper studies the effect of incorporating class-wise k-nearest neighbor  models in the input space to calibrate the prediction confidences from a given classifier, to address the “trustworthiness prediction” task: a narrower variant of confidence calibration or out-of-distribution (OOD) detection tasks, where the focus is to define a score function capable to discriminate whether a given (test) input will be correctly/incorrectly classified. The sufficient experiments verify that the proposed method works better than existing sota methods sota experiments. In general, the main idea of this paper and the theory proposed in this paper are interesting. However, the authors should merge all rebuttal information before the full acceptance. Thus, I would like to recommend accepting the submission with minor revision.

---

> ### Author Response · Authors · 2022-08-22
> **We thank all reviewers and the action editor for the valuable comments on our work.**
>
> We would like to thank the reviewers and the action editor for providing helpful suggestions and comments.
> We have included all the requested changes to this final version and added the code link to the abstract part.